# Association between the early use of beta-blocker and the risk of sepsis-associated acute kidney injury: A retrospective cohort study using the MIMIC-IV database

Canmin Wang◯°*, Yingfang Hu°, Yunfeng Song

Department of Intensive Care Unit, The Affiliated Guangdong Second Provincial General Hospital of Jinan University, Guangzhou, Guangdong Province, P.R. China

☺ These authors contributed equally to this work.
* wangcanMin08@163.com

## Abstract

### Background

Sepsis-associated acute kidney injury (SA-AKI) is a common and life-threatening complication in critically ill patients. Studies have shown that the use of beta-blockers improves hemodynamics and the risk of death in patients with sepsis. However, the association between beta-blockers use and the risk of AKI in patients with sepsis remains poorly understood. The present study aimed to evaluate this potential association.

### Method

Sepsis patients for this retrospective cohort study were extracted from the Medical Information Mart for Intensive Care-IV (MIMIC) database. Propensity score matching (PSM) was used to balance the basic characteristics between beta-blocker users and non-users. Univariate and multivariable logistic regression analysis were employed to evaluate the association between early use of beta-blocker and SA-AKI. Odds ratio (OR) and 95% confidence interval (CI) were estimated as effect measurements.

### Results

Totally 4,419 patients with sepsis were enrolled in our study. The follow-up period was from the 24th hour of intensive care unit (ICU) admission to the occurrence of AKI or ICU discharge, with 2,122 (48.02%) cases of developed AKI. After PSM, a lower SA-AKI risk was observed in the early use of the beta-blockers group compared to the non-user group (adjusted OR: 0.80; 95%CI: 0.64–0.99). Similar associations of early use of beta-blockers and SA-AKI were observed in patients younger than 65 years old, male, without comorbidities, and with Simplified Acute Physiology Score II/ Charlson comorbidity index scores below the median (all $P < 0.05$).

**Data availability statement:** The datasets generated during and/or analyzed during the current study are available in the MIMIC IV database, https://mimic.physionet.org/iv/.

**Funding:** The author(s) received no specific funding for this work.

**Competing interests:** The authors have declared that no competing interests exist.

**Abbreviations:** SA-AKI, sepsis-associated acute kidney injury; ICU, intensive care unit; AMI, acute myocardial infarction; MIMIC, Medical Information Mart for Intensive Care; eGFR, estimated glomerular filtration rate; RRT, renal replacement therapy; KDIGO, Kidney Disease Improving Global Outcomes; CKD, chronic kidney disease; SOFA, sequential organ failure assessment; CCI, Charlson comorbidity index; BUN, blood urea nitrogen; Cr, creatinine; WBC, white blood cell; RDW, red cell distribution width; INR, international normalized ratio; PT, prothrombin time; PTT, partial thromboplastin time; CABG, coronary artery bypass grafting; SD, standard deviation; OR: odds ratio; CI, confidence interval.

## Conclusion

In ICU patients with sepsis, early use of beta-blockers is associated with a reduced risk of AKI, which may help reduce renal impairment and improve survival. Further studies are needed to verify the underlying mechanisms of beta-blockers in the development of SA-AKI.

## Introduction

Sepsis is a life-threatening organ dysfunction caused by dysregulated systemic inflammation and immune response in response to infection [1]. Sepsis remains one of the leading causes of patient death due to the high morbidity and poor prognosis [2], causing more than $24 billion in healthcare-related costs each year in the United States [3]. Sepsis is often associated with many complications [4–6], among which sepsis-associated acute kidney injury (SA-AKI) is a life-threatening complication in critically ill patients [7]. Studies have reported that SA-AKI is the most common AKI syndrome in the intensive care unit (ICU), accounting for about half of all such AKI [8]. SA-AKI not only increases the mortality of patients with sepsis but also significantly prolongs the length of hospital stay and increases the medical cost [9].

By inhibiting beta-adrenergic receptor signaling, beta-blockers exert negative chronotropic and inotropic effects, thereby reducing cardiac workload and ameliorating myocardial ischemia. These mechanisms underpin their broad utility in managing cardiovascular diseases and associated comorbidities [10]. In addition to their well-established roles in hypertension, arrhythmia, and heart failure, emerging clinical evidence suggests that beta-blockers may confer renal protective benefits in patients with cardiovascular complications [11–14]. Studies have shown that the use of beta-blockers is associated with a lower risk of AKI in patients with acute myocardial infarction (AMI) [12,13], acute coronary syndrome [11], and atrial fibrillation [14]. This may be related to the antagonism of catecholamine adrenergic stimulation to prevent sympathetic hyperactivity and the improvement of endothelial dysfunction associated with renal ischemia [15]. However, the results of current studies are inconsistent. A study by Kim et al. reported that non-selective beta-blockers can also increase the risk of renal hypoperfusion and AKI in patients susceptible to cirrhosis by reducing cardiac output and blood pressure [16].

In sepsis patients, there is also evidence shown beta-blockers can improve hemodynamics [17] and reduce the risk of in-hospital mortality [18]. Generally speaking, the pathophysiology of SA-AKI is thought to be related to decreased renal blood flow and ischemia [9]. Currently, there is still a lack of research on the association between beta-blockers and the risk of SA-AKI. Therefore, this study aims to explore the association between early use of beta-blockers and the risk of AKI in ICU patients with sepsis. Our study may provide scientific guidance for the clinical management of sepsis patients and improve the prognosis and quality of life.

## Methods

### Sources of data

The Medical Information Mart for Intensive Care IV (MIMIC-IV) database (https://mimic.mit.edu/docs/iv/) is a publicly available, anonymous, research-oriented clinical database that collects patient data from the ICU. MIMIC-IV contains comprehensive information on each ICU patient's hospitalization, including demographic information, admission information, laboratory measurements, medications given, and vital signs. The rich data from the MIMIC database allows us to conduct a comprehensive assessment of the association between beta-blocker use and the risk of AKI in sepsis patients. The database received approval from the institutional review boards of the Massachusetts Institute of Technology (MIT, located in Cambridge, Massachusetts, USA) and Beth Israel Deaconess Medical Center (BIDMC). The data is publicly available (in the MIMIC-IV database), therefore, the ethical approval statement and the requirement for informed consent were waived for this study. This study adhered to the strengthening the reporting of observational studies in epidemiology (STROBE) guidelines and complied with the principles of the Declaration of Helsinki [19].

### Study population

This retrospective cohort study included 4,419 patients from the MIMIC IV database from 2008 to 2019. Inclusion criteria for patients were as follows: 1) aged 18 years or older; 2) first admission to the ICU; 3) diagnosis of sepsis according to the Sepsis-3 criteria [20]; 4) documented baseline use of beta-blockers (yes, no); and 5) available data on the occurrence of AKI. The exclusion criteria of patients were: 1) ICU stay of less than 24 hours; 2) baseline estimated glomerular filtration rate (eGFR) under 15 ml/min/1.73m²; 3) use of renal replacement therapy (RRT) at baseline; 4) presence of AKI at baseline; and 5) mortality within the first 24 hours of ICU admission.

### Beta-blockers exposure

The exposure of present study was the use of beta-blocker. Exposure to beta-blockers was defined as any form or dose of beta-blocker used within 24 hours after admission to the ICU. The beta-blockers included "Labetalol", "Metoprolol" and "Esmolol".

### Outcome of interest

The primary outcome was defined as the occurrence of AKI according to the Kidney Disease Improving Global Outcomes (KDIGO) [20]. AKI was diagnosed if any of the criteria were met: an elevation in serum creatinine (SCr) by ≥ 0.3 mg/dL within 48 hours; an increase to 1.5 times the baseline SCr level within the prior 7 days; or urine volume of less than 0.5 mL/kg/hour for 6 hours. Patients were followed from the first 24 hours after ICU admission until AKI occurrence or ICU discharge.

### Data collection

Data collection involved the utilization of Structured Query Language (SQL) with PostgreSQL (version 14.2) to extract baseline characteristics of patients with sepsis. Potential covariates influence the results were included in this study: age (age; continuous variables), gender (male, female), race (Black, White, other, unknown), weight (kg), heart failure (yes, no), AMI (yes, no) [21], chronic kidney disease (CKD; yes, no) [22], hypertension (yes, no), diabetes (yes, no), heart rate (bpm), systolic (mmHg), diastolic (mmHg), respiratory rate (bpm), temperature (°C), saturation of peripheral oxygen ($SpO_2$; %), sequential organ failure assessment (SOFA; score), Simplified Acute Physiology Score II (SAPS II; score), Charlson comorbidity index (CCI; score), creatinine (Cr; mg/mL), blood urea nitrogen (BUN; mg/dL), platelet (K/uL), white blood cell (BC, K/uL), red cell distribution width (RDW, %), hemoglobin

(g/dL), hematocrit (%), glucose (mg/dL), calcium (mmol/L), bicarbonate (mEq/L), sodium (mEq/L), potassium (mEq/L), chloride (mEq/L), international normalized ratio (INR; ratio), prothrombin time (PT; second), partial thromboplastin time (PTT; second), 24-hour urine-output (mL), ventilation (yes, no), vasopressor (yes, no), loop diuretics (yes, no), nephrotoxic antibiotics (yes, no), coronary artery bypass grafting (CABG; yes, no), insulin (yes, no), platelet infusion (yes, no); eGFR (ml/min/1.73m$^2$). The time of laboratory test was the first record between ICU admission and 24$^{th}$ hours of ICU admission.

## Statistical analysis

For continuous data, the normality was assessed by skewness and kurtosis, while the homogeneity of variance was assessed by Levene's test. Normally distributed variables were presented as mean ± standard deviation (SD) and compared using the Student's t-test for equal variances or the Satterthwaite t-test for unequal variances. Non-normal data were described by the median and interquartile range [M (Q$_1$, Q$_3$)], and the Wilcoxon rank sum test was used for comparison between groups. Categorical variables were expressed by frequencies and percentages n (%) and the comparison between groups was performed by chi-square test or Fisher's exact test. If the proportion of missing data was more than 20%, the variable was deleted; otherwise, the random forest imputation method was utilized, as shown in Table S1 in S1 File.

To balance baseline characteristics, the nearest neighbor matching method was used to match the early use of beta-blockers with a 1:1 (yes, no) no-return matching, and the caliper width was set to 0.018. Table 1 shows the balance of basic characteristics before and after PSM. The standardized mean difference (SMD) < 0.1 was considered an acceptable threshold of PSM. To screen confounding factors, univariate and multivariable logistic regression models were used to assess the effect of potential covariates on the risk of AKI in sepsis patients. Model 1 adjusted for none and Model 2 adjusted for all confounding factors retained by two-way stepwise regression in the previous step. The odds ratio (OR) and the 95% confidence interval (CI) of the association between beta-blockers and the risk of AKI were assessed by the logistic regression model. Subgroup analysis was performed according to age (< 65, ≥ 65), gender (male, female), SAPS II (< 34, ≥ 34), CCI (< 2, ≥ 2), and comorbidities (heart failure, AMI, CKD, diabetes). A two-sided $P$ value <0.05 was considered statistically significant. Data cleaning and missing data imputation were performed with Python (version 3.9.12). Sensitivity analyses, difference comparisons, statistical modeling, and plotting were performed with R (version 4.3.1, 2023-06-16 ucrt).

## Results

### Basic characteristics of participants

In total, 4,419 sepsis patients were enrolled in this cohort study, of whom 2,122 (48.02%) cases developed AKI (Fig 1). Table 1 summarizes the basic characteristics of patients. There were 750 patients (16.97%) exposed to beta-blockers within 24 hours of ICU admission, and 3,669 (83.03) were not. The mean age of patients was 62.17 ± 16.65 years old, with most cases being male (61.64%). Patients were divided into two groups based on the early use of beta-blockers. Before PSM, the differences were observed in ICU duration, age, heart failure, CKD, hypertension, diabetes, systolic, diastolic, respiratory rate, SpO$_2$, SOFA, SAPS II, CCI, Cr, BUN, platelet, RDW, glucose, calcium, PT, PTT, 24-hour urine-output, ventilation, vasopressor use, loop diuretics use, nephrotoxic antibiotics use, insulin use and eGFR (all $P$ < 0.05). After PSM, 1,452 patients were successfully matched. The distribution of propensity scores close similarity between groups (**Supplementary Figure 1A** in S1 File) and the covariate balance improved (all SMD < 0.1) through PSM (**Supplementary Figure 1B** in S1 File). No significant differences were found between groups for all variables except glucose ($P$ < 0.05).

**Table 1. Characteristics of the study population.**

| Variables | Unmatched patients | | | Propensity scores matched patients | | |
|---|---|---|---|---|---|---|
| | Early beta-blockers users (N=3669) | Non-users (N=750) | P | Early beta-blockers users (N=3669) | Non-users (N=726) | P |
| ICU duration, day, M (Q₁, Q₃) | 2.97 (1.78-5.36) | 2.72 (1.43-5.5) | 0.005 | 2.64 (1.61-4.89) | 2.69 (1.41-5.6) | 0.910 |
| Age, years, Mean±SD | 61.62±16.92 | 64.83±14.96 | <0.001 | 65.18±14.83 | 64.53±14.86 | 0.410 |
| Gender, n (%) | | | 0.330 | | | 0.330 |
| Female | 1395 (38.02) | 300 (40) | | 265 (36.5) | 284 (39.12) | |
| Male | 2274 (61.98) | 450 (60) | | 461 (63.5) | 442 (60.88) | |
| Race, n (%) | | | 0.099 | | | 0.574 |
| Black | 229 (6.24) | 61 (8.13) | | 48 (6.61) | 57 (7.85) | |
| Others | 524 (14.28) | 118 (15.73) | | 102 (14.05) | 112 (15.43) | |
| Unknown | 465 (12.67) | 81 (10.8) | | 72 (9.92) | 77 (10.61) | |
| White | 2451 (66.8) | 490 (65.33) | | 504 (69.42) | 480 (66.12) | |
| Weight, kg, Mean±SD | 77.82±17.05 | 79.07±18.03 | 0.080 | 79.20±17.69 | 79.07±17.93 | 0.892 |
| Heart failure, n (%) | | | 0.006 | | | 0.593 |
| No | 2777 (75.69) | 603 (80.4) | | 592 (81.54) | 583 (80.3) | |
| Yes | 892 (24.31) | 147 (19.6) | | 134 (18.46) | 143 (19.7) | |
| AMI, n (%) | | | 0.109 | | | 0.768 |
| No | 3303 (90.02) | 690 (92) | | 672 (92.56) | 668 (92.01) | |
| Yes | 366 (9.98) | 60 (8) | | 54 (7.44) | 58 (7.99) | |
| CKD, n (%) | | | 0.001 | | | 1.000 |
| No | 3157 (86.05) | 681 (90.8) | | 658 (90.63) | 658 (90.63) | |
| Yes | 512 (13.95) | 69 (9.2) | | 68 (9.37) | 68 (9.37) | |
| Hypertension, n (%) | | | <0.001 | | | 0.653 |
| No | 1707 (46.52) | 241 (32.13) | | 230 (31.68) | 239 (32.92) | |
| Yes | 1962 (53.48) | 509 (67.87) | | 496 (68.32) | 487 (67.08) | |
| Diabetes, n (%) | | | 0.001 | | | 0.644 |
| No | 2768 (75.44) | 523 (69.73) | | 520 (71.63) | 511 (70.39) | |
| Yes | 901 (24.56) | 227 (30.27) | | 206 (28.37) | 215 (29.61) | |
| Heart rate, bpm, Mean±SD | 87.31±18.53 | 89.97±19.15 | <0.001 | 89.91±17.66 | 89.54±18.95 | 0.703 |
| Systolic, mmHg, Mean±SD | 122.26±22.76 | 128.74±25.04 | <0.001 | 127.32±23.12 | 127.97±24.47 | 0.607 |
| Diastolic, mmHg, Mean±SD | 66.80±15.63 | 68.48±15.67 | 0.007 | 67.62±15.28 | 68.25±15.62 | 0.435 |
| Respiratory rate, bpm, Mean±SD | 18.41±5.67 | 17.95±5.79 | 0.042 | 17.68±5.76 | 17.82±5.74 | 0.634 |
| Temperature, °C, Mean ±SD | 36.64±0.71 | 36.62±0.71 | 0.320 | 36.63±0.74 | 36.61±0.70 | 0.704 |
| SpO₂, %, Mean±SD | 97.68±2.92 | 98.07±2.77 | <0.001 | 97.92±2.87 | 98.04±2.79 | 0.420 |
| SOFA, score, Mean±SD | 5.27±2.87 | 4.75±2.44 | <0.001 | 4.75±2.48 | 4.77±2.44 | 0.915 |
| SAPS II, score, Mean±SD | 33.86±11.74 | 34.98±11.34 | 0.016 | 35.01±11.97 | 34.83±11.30 | 0.763 |
| CCI, score, Mean±SD | 2.50±2.32 | 2.35±2.21 | 0.103 | 2.37±2.26 | 2.37±2.22 | 0.944 |
| Creatinine, mg/dL, Mean±SD | 1.01 (±0.56) | 0.91 (±0.45) | <0.001 | 0.8 (0.6-1) | 0.8 (0.6-1) | 0.889 |
| BUN, mg/dL, M (Q₁, Q₃) | 17 (12-25) | 16 (12-22) | 0.004 | 16 (12-22) | 16 (12-22) | 0.952 |
| Platelet, K/uL, Mean±SD | 198.20±103.65 | 207.66±104.75 | 0.023 | 206.42±105.35 | 206.80±105.02 | 0.945 |
| WBC, K/uL, M (Q₁, Q₃) | 11.4 (8.1-15.3) | 11.45 (8.8-15) | 0.227 | 11.9 (8.5-15.5) | 11.5 (8.9-15) | 0.925 |
| RDW, %, Mean±SD | 14.73±2.19 | 14.49±2.01 | 0.003 | 14.46±1.94 | 14.49±2.02 | 0.794 |
| Hemoglobin, g/dL, Mean±SD | 10.63±2.20 | 10.59±2.24 | 0.710 | 10.61±2.16 | 10.61±2.25 | 0.994 |
| Hematocrit, %, Mean±SD | 31.96±6.47 | 31.82±6.57 | 0.602 | 31.86±6.30 | 31.87±6.58 | 0.982 |
| Glucose, mg/dL, M (Q₁, Q₃) | 128 (106-160) | 135.5 (114-162) | <0.001 | 128.5 (108-159.75) | 135 (113-162) | 0.024 |
| Calcium, mmol/L, Mean ±SD | 8.24±0.76 | 8.31±0.68 | 0.010 | 8.31±0.72 | 8.31±0.67 | 0.999 |

*(Continued)*

**Table 1.** (Continued)

| Variables | Unmatched patients | | | Propensity scores matched patients | | |
|---|---|---|---|---|---|---|
| | Early beta-blockers users (N = 3669) | Non-users (N = 750) | P | Early beta-blockers users (N = 3669) | Non-users (N = 726) | P |
| Bicarbonate, mEq/L, Mean±SD | 23.59±4.27 | 23.67±3.70 | 0.610 | 23.73±3.81 | 23.69±3.69 | 0.833 |
| Sodium, mEq/L, Mean±SD | 137.90±4.77 | 137.70±4.35 | 0.271 | 137.65±4.95 | 137.66±4.32 | 0.964 |
| Potassium, mEq/L, Mean±SD | 4.24±0.78 | 4.24±0.75 | 0.870 | 4.27±0.79 | 4.24±0.76 | 0.501 |
| Chloride, mEq/L, Mean±SD | 104.79±6.01 | 105.07±5.13 | 0.183 | 105.04±5.82 | 105.02±5.11 | 0.950 |
| INR, ratio, M ($Q_1$, $Q_3$) | 1.29 (1.17-1.5) | 1.28 (1.1-1.4) | 0.054 | 1.27 (1.1-1.5) | 1.28 (1.1-1.4) | 0.469 |
| PT, second, M ($Q_1$, $Q_3$) | 14.37 (12.9-16.1) | 14.3 (12.9-15.6) | 0.045 | 14.36 (12.72-16) | 14.3 (12.9-15.6) | 0.256 |
| PTT, second, Mean (±SD) | 32.54±7.42 | 31.83±7.13 | 0.013 | 31.92±6.98 | 31.94±7.17 | 0.965 |
| 24-hour urine-output, mL, M ($Q_1$, $Q_3$) | 2241 (1645-3000) | 2430 (1765.75-3255) | <0.001 | 2557.40 (±1228.57) | 2598.80 (±1218.24) | 0.519 |
| Ventilation, n (%) | | | <0.001 | | | 0.646 |
| No | 527 (14.36) | 63 (8.4) | | 68 (9.37) | 62 (8.54) | |
| Yes | 3142 (85.64) | 687 (91.6) | | 658 (90.63) | 664 (91.46) | |
| Vasopressor, n (%) | | | 0.031 | | | 0.519 |
| No | 2125 (57.92) | 467 (62.27) | | 434 (59.78) | 447 (61.57) | |
| Yes | 1544 (42.08) | 283 (37.73) | | 292 (40.22) | 279 (38.43) | |
| Loop diuretics, n (%) | | | <0.001 | | | 1.000 |
| No | 2760 (75.22) | 512 (68.27) | | 497 (68.46) | 498 (68.6) | |
| Yes | 909 (24.78) | 238 (31.73) | | 229 (31.54) | 228 (31.4) | |
| Nephrotoxic antibiotics, n (%) | | | 0.032 | | | 0.212 |
| No | 2196 (59.85) | 481 (64.13) | | 443 (61.02) | 467 (64.33) | |
| Yes | 1473 (40.15) | 269 (35.87) | | 283 (38.98) | 259 (35.67) | |
| CABG, n (%) | | | 0.212 | | | 0.500 |
| No | 3642 (99.26) | 748 (99.73) | | 726 (100) | 724 (99.72) | |
| Yes | 27 (0.74) | 2 (0.27) | | 0 (0) | 2 (0.28) | |
| Insulin, n (%) | | | <0.001 | | | 0.829 |
| No | 1963 (53.5) | 290 (38.67) | | 279 (38.43) | 284 (39.12) | |
| Yes | 1706 (46.5) | 460 (61.33) | | 447 (61.57) | 442 (60.88) | |
| Platelet infusion, n (%) | | | 0.736 | | | 0.813 |
| No | 3474 (94.69) | 713 (95.07) | | 687 (94.63) | 690 (95.04) | |
| Yes | 195 (5.31) | 37 (4.93) | | 39 (5.37) | 36 (4.96) | |
| eGFR, ml/min/1.73m$^2$, Mean±SD | 86.88±29.94 | 89.18±25.28 | 0.028 | 88.84±26.10 | 89.32±25.29 | 0.727 |
| AKI, n (%) | | | 0.350 | | | 0.052 |
| No | 1895 (51.65) | 402 (53.6) | | 353 (48.62) | 391 (53.86) | |
| Yes | 1774 (48.35) | 348 (46.4) | | 373 (51.38) | 335 (46.14) | |

ICU, intensive care unit; M ($Q_1$, $Q_3$), median (first Quartile, third Quartile); SD, standard deviation; n (%), number (percentage); AMI, acute myocardial infarction; CKD, chronic kidney disease; $SpO_2$, saturation of peripheral oxygen; SOFA, sequential organ failure assessment; SAPS II, Simplified Acute Physiology Score II; CCI, Charlson comorbidity index; BUN, blood urea nitrogen; WBC, white blood cell; RDW, red cell distribution width; INR, international normalized ratio; PT, prothrombin time; PTT, partial thromboplastin time; CABG, coronary artery bypass grafting; eGFR, estimated glomerular filtration rate; AKI, acute kidney injury. P-values were obtained using the Student's t test for normal continuous variables or Satterthwaite t test, the Wilcoxon rank sum test for non-normal continuous variables, and the Chi-square test or Fisher's exact test for categorical variables.

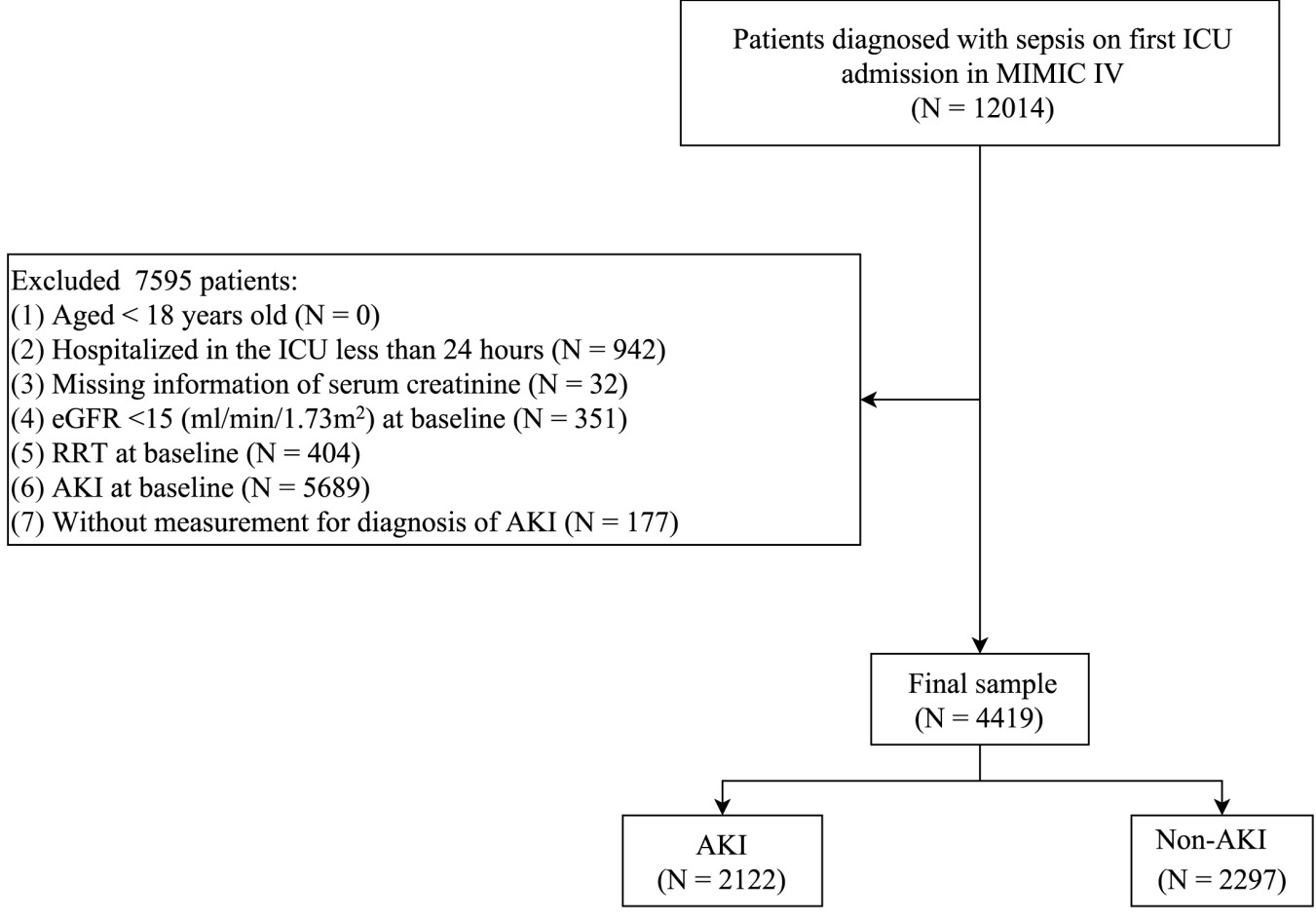

**Fig 1. Flow diagram of selecting eligible participants.** ICU, intensive care unit; MIMIC, Medical Information Mart for Intensive Care; eGFR, estimated glomerular filtration rate; RRT, renal replacement therapy; AKI, acute kidney injury.

## Association of early use of beta-blocker with SA-AKI

As can be seen in Table 2, the unadjusted model (model 1) shows early use of beta-blockers was associated with a lower risk of SA-AKI (OR: 0.81; 95%CI: 0.66–1.00, P = 0.046). To exclude the influence of confounding factors, the covariate screening was presented in Table S2 in S1 File. When adjusting for age, weight, heart failure, systolic, temperature, SOFA, BUN, WBC, chloride, loop diuretics use, and eGFR in model 2, a similar association of early use of beta-blocker and SA-AKI was found with an OR value of 0.80 (95%CI: 0.64–0.99, P = 0.042). The results of covariate screening before PSM are shown in Table S3 in S1 File. No association was observed between early beta-blocker use and SA-AKI in either model 1 (adjusted for none) or model 2 (adjusted for covariates) before PSM.

## Association of early use of beta-blocker with SA-AKI in different subgroups

To assess the association of early use of beta-blockers with SA-AKI, patients were grouped by age (<65; ≥65), gender, baseline SAPSII, CCI, and comorbidities. As shown in Fig 2, early use of beta-blockers was associated with a lower risk of developing AKI in sepsis patients younger than 65 years old, male, without comorbidities (heart failure, AMI, CKD, diabetes), and with SAPS II/CCI scores below the median (all *P* < 0.05).

**Table 2. Association between early use of beta-blockers and SA-AKI before and after PSM.**

| Variables | Model 1 | | Model 2 | |
|---|---|---|---|---|
| | OR (95% CI) | *P* | OR (95% CI) | *P* |
| **Before PSM** | | | | |
| Beta-blockers | | | | |
| No | Ref | | Ref | |
| Yes | 0.92 (0.79-1.08) | 0.330 | 0.96 (0.81-1.14) | 0.627 |
| **After PSM** | | | | |
| Beta-blockers | | | | |
| No | Ref | | Ref | |
| Yes | 0.81 (0.66-1.00) | 0.046 | 0.80 (0.64-0.99) | 0.042 |

SA-AKI, sepsis-associated acute kidney injury; PSM, propensity score matching; OR: odds ratio; CI: confidence intervals; PSM, propensity score matching; Ref: reference; SOFA, sequential organ failure assessment; SAPS II, Simplified Acute Physiology Score II; WBC, white blood cell; PTT, partial thromboplastin time; CABG, coronary artery bypass grafting; BUN, blood urea nitrogen; eGFR, estimated glomerular filtration rate.

Model 1 adjusted for none. Before PSM, Model 2 adjusted for age, weight, heart failure, AMI, respiratory rate, SOFA, SAPS II, creatinine, WBC, chloride, PTT, 24-hour urine-output; ventilation, loop diuretics, nephrotoxic antibiotics, and CABG. After PSM, Model 2 adjusted for age, weight, heart failure, systolic, temperature, SOFA, BUN, WBC, chloride, loop diuretics, and eGFR.

### Association between the types of beta-blockers with SA-AKI

To further explore the association between the types of beta-blockers and the risk of SA-AKI, covariates were screened in the population who used beta-blockers early (Table S4 in S1 File). No association between the type of beta-blockers used in the early stage and the risk of SA-AKI was observed (all *P* > 0.05) (Table 3).

### Discussion

In this retrospective cohort study, early use of beta-blockers after admission to ICU was found to be associated with a lower risk of AKI in sepsis patients. Similarly, early use of beta-blockers was associated with a lower risk of AKI in sepsis patients of male gender, aged < 65 years old, without comorbidities, and with SAPS II/CCI scores below the median. Further using Labetalol as a reference, no difference in SA-AKI risk was found between groups of Metoprolol, Esmolol, and those using More than two types of beta-blockers. This finding provides support for the potential clinical benefit of beta-blockers in the treatment of sepsis.

The association between beta-blockers and the risk of AKI has been the focus of research. A prospective observational study of 1,309 AMI patients undergoing coronary angiography (CAG) or percutaneous coronary (PCI) intervention found that use of beta-blockers within 24 hours perioperatively may be associated with reduced rates of contrast-induced AKI and long-term mortality in AMI patients undergoing CAG or PCI [13]. In another study involving 4,527 patients hospitalized for atrial fibrillation, administration of a beta-blocker before admission was found to be associated with a reduced risk of in-hospital AKI [14]. Leung et al. [11] conducted a cohort study of 5,911 participants (aged ≥ 66 years) with acute coronary syndromes in Alberta, who underwent coronary angiography and found that patients with stage 2–3 AKI had 54% lower odds of using beta-blockers compared with the non-AKI group. However, studies on the beta-blockers and AKI in sepsis patients are still lacking. Our study provides evidence that beta-blockers in septic ICU patients may help to reduce the risk of SA-AKI in septic patients.

Sepsis is a potentially life-threatening multiple-organ dysfunction, and approximately 60% of sepsis patients suffer from AKI [23]. The pathogenesis of sepsis-induced AKI is still not fully clarified but is likely to be associated with a dysregulated immune response and systemic inflammation, hemodynamic changes, and renal microvascular endothelial cell dysfunction [24]. The demonstrated benefits of beta-blockers in improving heart rate and blood pressure, reducing mortality and

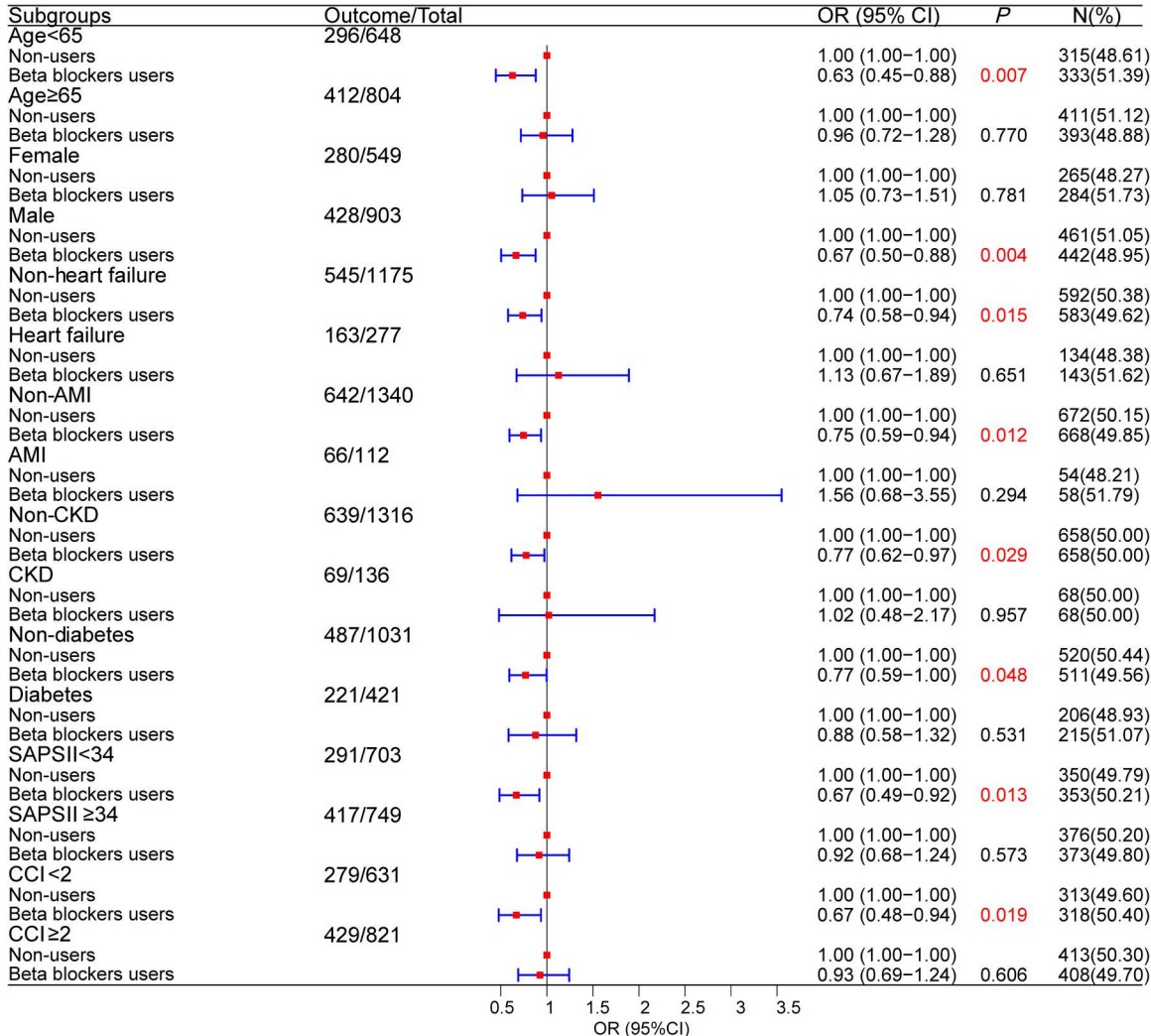

**Fig 2. Association of early use of beta-blocker with SA-AKI in different subgroups.** SA-AKI, sepsis-associated acute kidney injury; AMI, acute myocardial infarction; CKD, chronic kidney disease; SAPS II, Simplified Acute Physiology Score II; CCI, Charlson comorbidity index.

**Table 3. Association between the type of beta-blockers and SA-AKI after PSM.**

| Variates | OR (95%CI) | P | OR (95%CI) | P | OR (95%CI) | P |
|---|---|---|---|---|---|---|
| Beta blockers | | | | | | |
| Labetalol | 0.47 (0.15-1.53) | 0.212 | Ref | | 1.40 (0.87-2.25) | 0.160 |
| Metoprolol | 0.34 (0.11-1.02) | 0.054 | 0.71 (0.45-1.14) | 0.160 | Ref | |
| Esmolol | Ref | | 2.11 (0.65-6.81) | 0.212 | 2.95 (0.98-8.91) | 0.054 |

SA-AKI, sepsis-associated acute kidney injury; PSM, propensity score matching; OR: odds ratio; CI: confidence intervals; Ref: reference; SOFA, sequential organ failure assessment; BUN, blood urea nitrogen; WBC, white blood cell; eGFR, estimated glomerular filtration rate. Model 1 adjusted for none. Model 2 adjusted for age, weight, heart failure, systolic, temperature, SOFA, BUN, WBC, Chloride; loop diuretics; eGFR.

length of hospital stay in sepsis, and septic shock make beta-blockers a potential therapy for the treatment of sepsis patients [25]. The association between early use of beta-blockers and SA-AKI was observed in our study, but the underlying mechanisms remain unclear. Firstly, in severe pathological states such as sepsis, the sympathetic nervous system is often overexcited, leading to systemic vasoconstriction and increased heart rate. Beta-blockers prevent sympathetic hyperactivity by antagonizing catecholamine adrenergic stimulation, reducing heart rate and blood pressure, and facilitating the maintenance of renal blood flow [15]. Second, beta-blockers also ameliorate renal ischemia endothelial dysfunction through endothelial nitric oxide synthase (eNOS) activation [26]. Elevated plasma endothelin-1 (ET-1) levels have been reported in patients with sepsis [27]. Due to the potent renal vasoconstrictor effect of ET-1, which may be associated with acute renal failure [28]. ET-1 is normally induced by hypoxia-inducible factor 1-alpha (HIF-1α) [29], downregulated by nitric oxide (NO) [28]. Ogura et al. [29] conducted an experiment with a rat model of endotoxemia and showed that landiolol hydrochloride ameliorated the up-regulation of the HIF-1α-ET-1 system in the kidneys with minimal morphological damage and normalized biomarkers of early renal injury in this rat model of endotoxemia. Third, the benefit of beta-blockers on SA-AKI may be related to their role in regulating the inflammatory response. An animal study reported reduced serum interleukin-6 (IL-6), high mobility group protein B1 (HMGB)-1, and tumor necrosis factor-α(TNF-α) levels but increased interleukin-10 (IL-10) levels in the group given esmolol compared with the sepsis group [30]. Another study also reported that atenolol increased IL-10 release in septic sheep [31]. IL-10 is an important immunomodulatory cytokine, which inhibits and terminates the inflammatory immune response by inhibiting the activation of monocytes and macrophages [32]. Abdel Kawy [33] studied the effects of carvedilol, a beta-adrenoceptor antagonist with antioxidant activity, on septic kidney injury induced by cecal ligation and puncture (CLP) in rats and found that carvedilol attenuated CLP-induced intracellular renal edema and inflammation, underlining the renoprotective role of carvedilol in sepsis. In summary, sepsis leads to blood flow redistribution and microcirculation disturbance through complex molecular mechanisms (sympathetic nervous system activation, endothelial injury, and inflammation), which further damages renal tissue and leads to AKI. Beta-blockers may reduce the risk of SA-AKI through these mechanisms.

Our results demonstrated a benefit of early use of beta-blockers in males. It is likely that estrogen inhibits beta-adrenoceptor cardiac expression and reduces beta-adrenoceptor mediated stimulation [34]. Similar benefits of beta-blockers can be found in patients aged < 65 years, without comorbidities, and with SAPS II/CCI scores below the median. This effect may be attributed to greater cardiac reserve in these individuals, allowing better tolerance to the heart rate reduction and blood pressure lowering effects of beta-blockers. Consequently, they are more likely to experience the cardioprotective benefits without exacerbating renal hypoperfusion from cardiac insufficiency. Additionally, these patients typically exhibit enhanced renal blood flow regulation, enabling them to adapt to the hemodynamic changes induced by beta-blockers and maintain better glomerular filtration rates.

We also found that in our study, the association between early beta-blockers use and the risk of AKI in sepsis patients was only observed after PSM. We attempted to explain the possible reasons for the inconsistency in the association between exposure and outcome before and after PSM. First, before PSM, significant differences existed between beta-blocker users and non-users in demographic characteristics and comorbidities. These confounding factors may be associated with both beta-blockers use and the risk of AKI in patients with sepsis but exert opposing effects, thereby obscuring the true association between the early beta-blocker use and AKI in this population. Second, the unmatched raw sample may have been subject to non-random allocation, where clinicians were more likely to prescribe beta-blockers to specific subgroups of sepsis patients, thereby introducing selection bias into the sample. Moreover, before PSM, the sample sizes between groups were imbalanced, with significantly fewer patients in the early beta-blocker use group compared to the non-use group (3,669 vs. 750), resulting in insufficient statistical power. Potential explanations for the statistically significant association between early beta-blocker use and the risk of AKI in patients with sepsis after PSM include: first, PSM attenuated the interference of confounding factors on the exposure-outcome relationship by balancing covariates between the two groups (such as age, comorbidities, SOFA score, renal function parameters, and concomitant medications), thereby unmasking the inherent

protective effect of beta-blockers; second, during the PSM process, extreme "unmatchable" cases were excluded, as these patients might have otherwise confounded the assessment of the association between early beta-blockers use and the risk of AKI in sepsis patients; moreover, as has been explained above, there are already clinical studies related to physiological mechanisms that support the beneficial role of beta-blockers in maintaining renal function [26–30].

This study also stratified results by type of beta-blocker. When compared to Labetalol, Metoprolol, Esmolol, or combinations of more than two types did not show the difference in the risk of SA-AKI. This suggests that the mechanisms of action for different beta-blockers on SA-AKI may be similar. Therefore, the clinical selection of beta-blockers should consider the specific conditions of individual patients, as well as the side effects, onset time, and accessibility of the drug.

Our study is the first study to explore the association between beta-blockers and the risk of AKI in patients with sepsis, which provides a basis for drug decision-making in patients at high risk of poor prognosis. Nonetheless, several important limitations of this study must be acknowledged. First, inherent to the retrospective cohort design, our study was susceptible to recall bias and potential incompleteness of medical records. While we adjusted for multiple known confounders in our logistic regression models, residual confounding from unmeasured variables may persist. Importantly, our findings demonstrate an association between β-blocker use and AKI risk in septic patients but cannot establish causality Second, as our data were obtained exclusively from the MIMIC database representing a single academic medical center, the generalizability of our findings to other populations with different demographic characteristics, healthcare systems, or practice patterns may be limited. Furthermore, the ICU-based nature of this dataset restricts the applicability of our results to non-critical care settings such as general wards or outpatient populations. Third, as an observational study, the exposure factors were not randomized. Although PSM was employed to minimize bias between beta-blocker groups, larger prospective randomized controlled trials are needed. Fourth, due to the availability of database, we were unable to stratify patients by infection site. Consequently, we could not assess whether the therapeutic effects of beta-blockers varied across different sources of sepsis. Additionally, as the MIMIC-IV database lacks records of beta-blocker dosage information, we were also unable to determine whether the association between early beta-blocker use and the risk of AKI in sepsis patients varied according to dosage differences. Future large-scale, well-designed multicenter prospective cohort studies or randomized controlled trials (RCTs) are warranted to further validate the association between β-blocker use and AKI risk in septic patients, while mechanistic animal studies are needed to elucidate the underlying biological pathways involved.

## Conclusion

In ICU patients with sepsis, early use of beta-blockers is associated with a reduced risk of AKI, which can provide a reference for improving the prognosis of patients with sepsis. However, further studies are needed to investigate the mechanism of beta-blockers on SA-AKI.

## Supporting information

**S1 File. Table S1 Sensitivity analysis before and after imputation.** Table S2 The association between confounding variables and SA-AKI after PSM. Table S3 The association between confounding variables and SA-AKI before PSM. Table S4 The association between confounding variables and SA-AKI after PSM in early use of beta-blockers population. Figure 1A Distribution of propensity scores. Figure 1B Standardized mean differences before and after propensity score matching.
(ZIP)

## Author contributions

**Conceptualization:** Canmin Wang.

**Data curation:** Yingfang Hu, Yunfeng Song.

**Formal analysis:** Yingfang Hu, Yunfeng Song.

**Investigation:** Yingfang Hu, Yunfeng Song.

**Methodology:** Yingfang Hu, Yunfeng Song.

**Writing – original draft:** Canmin Wang.

**Writing – review & editing:** Canmin Wang.

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
