## [Decision Letter · Decision Letter 0]

PONE-D-24-60039Association between the early use of beta-blocker and the risk of sepsis-associated acute kidney injury: A retrospective cohort study using the MIMIC-IV databasePLOS ONE

Dear Dr. Wang,

Thank you for submitting your manuscript to PLOS ONE. After careful consideration, we feel that it has merit but does not fully meet PLOS ONE’s publication criteria as it currently stands. Therefore, we invite you to submit a revised version of the manuscript that addresses the points raised during the review process.

We look forward to receiving your revised manuscript.

Kind regards,

Amirmohammad Khalaji

Academic Editor

PLOS ONE

Journal Requirements:

Reviewers' comments:

Reviewer's Responses to Questions

**Comments to the Author**

1. Is the manuscript technically sound, and do the data support the conclusions?

Reviewer #1: Yes

Reviewer #2: Partly

2. Has the statistical analysis been performed appropriately and rigorously? 

Reviewer #1: I Don't Know

Reviewer #2: I Don't Know

3. Have the authors made all data underlying the findings in their manuscript fully available?

Reviewer #1: Yes

Reviewer #2: Yes

4. Is the manuscript presented in an intelligible fashion and written in standard English?

Reviewer #1: Yes

Reviewer #2: Yes

5. Review Comments to the Author

Reviewer #1: - Mention the aim of study in the abstract.

- The sentence " potential benefits of beta-blockers in the development of AKI" is akward. consider revising it.

- add proper citation for MIMIC database.

- List all beta-blockers in the associated method section.

- add proper reference for data collection related to the exposures, outcomes, and covariates.

- it is suggested to use p-value instead of confidence alpha level for explainingg the cut-off.

- it is suggested to relocate table 1.

- other limitations such as retrospective design, site of infection.

- is it possible to compare all types of beta-blockers with each other? in the current format, all of them were compared to labetolol.

Reviewer #2: Your title is mostly clear but needs minor grammatical and stylistic refinements for correctness and readability. Here is the corrected version:

“Association Between Early Beta-Blocker Use and the Risk of Sepsis-Associated Acute Kidney Injury: A Retrospective Cohort Study Using the MIMIC-IV Database”

Method section:

The statement about the Helsinki Declaration is unclear. You should say according to.

The definition of “baseline beta-blocker use” should be more explicit. Were beta-blockers initiated prior to ICU admission or only during ICU stay?

The definition of beta-blocker use (any dose within 24 hours) may lead to misclassification bias. Did the study assess:

Duration of use? (Was it only a single dose or ongoing treatment?)

Dosage variations? (Could different doses have different effects?)

Different types of beta-blockers? (Selective vs. non-selective beta-blockers)

The caliper width is set at 0.018. What is the rationale for this specific choice?

Unclear Direction of Association Before PSM vs. After PSM:

Before PSM, no significant association was found between early beta-blocker use and SA-AKI.

After PSM, early beta-blocker use was associated with a lower risk of SA-AKI.

However, the interpretation of this shift is missing. The authors should discuss possible reasons why the association appears only after matching.

6. PLOS authors have the option to publish the peer review history of their article (what does this mean? ). If published, this will include your full peer review and any attached files.

**Do you want your identity to be public for this peer review?** For information about this choice, including consent withdrawal, please see our Privacy Policy .

Reviewer #1: No

Reviewer #2: No

---

## [Author Response · Author response to Decision Letter 1]

1 May 2025

Comments to the Author

1. Is the manuscript technically sound, and do the data support the conclusions?

Reviewer #1: Yes

Reviewer #2: Partly

Reply: Thanks for your comments.

2. Has the statistical analysis been performed appropriately and rigorously?

Reviewer #1: I Don't Know

Reviewer #2: I Don't Know

Reply: Thanks for your comments.

3. Have the authors made all data underlying the findings in their manuscript fully available?

Reviewer #1: Yes

Reviewer #2: Yes

Reply: Thanks for your comments.

4. Is the manuscript presented in an intelligible fashion and written in standard English?

Reviewer #1: Yes

Reviewer #2: Yes

Reply: Thanks for your comments.

5. Review Comments to the Author

Reviewer #1: - Mention the aim of study in the abstract.

Reply: Thanks for your comments. We have mentioned the aim of this study in the Abstract: However, the association between beta-blockers use and the risk of AKI in patients with sepsis remains poorly understood. The present study aimed to evaluate this potential association (line 20-line 22).

- The sentence " potential benefits of beta-blockers in the development of AKI" is akward. consider revising it.

Reply: Thanks for your comments. We have revised this statement: By inhibiting beta-adrenergic receptor signaling, beta-blockers exert negative chronotropic and inotropic effects, thereby reducing cardiac workload and ameliorating myocardial ischemia. These mechanisms underpin their broad utility in managing cardiovascular diseases and associated comorbidities. In addition to their well-established roles in hypertension, arrhythmia, and heart failure, emerging clinical evidence suggests that beta-blockers may confer renal protective benefits in patients with cardiovascular complications (line 56-line 62).

- add proper citation for MIMIC database.

Reply: Thanks for your comments. We have provided the data source of the MIMIC-IV database in the manuscript: The Medical Information Mart for Intensive Care IV (MIMIC-IV) database (https://mimic.mit.edu/docs/iv/) is a publicly available, anonymous, research-oriented clinical database that collects patient data from the ICU (line 82-line 84).

- List all beta-blockers in the associated method section.

Reply: Thanks for your comment. We have listed all beta-blockers in the Beta-blockers exposure: The exposure of present study was the use of beta-blocker. Exposure to beta-blockers was defined as any form or dose of beta-blocker used within 24 hours of admission. The beta-blockers included “Labetalol”, “Metoprolol” and “Esmolol” (line 108-line 111).

- add proper reference for data collection related to the exposures, outcomes, and covariates.

Reply: Thanks for your comment. The exposures of present study were the use of beta-blockers. The endpoint was the incidence of AKI in patients with sepsis. AKI was defined in accordance with the Kidney Disease: Improving Global Outcomes (KDIGO) guidelines [Kellum, John A et al. “Diagnosis, evaluation, and management of acute kidney injury: a KDIGO summary (Part 1).” Critical care (London, England) vol. 17,1 204. 4 Feb. 2013, doi:10.1186/cc11454].

AKI was diagnosed if any of the criteria were met: an elevation in serum creatinine (SCr) by ≥ 0.3 mg/dL within 48 hours; an increase to 1.5 times the baseline SCr level within the prior 7 days; or urine volume of less than 0.5 mL/kg/hour for 6 hours. Patients were followed from the first 24 hours after ICU admission until AKI occurrence or ICU discharge.

Moreover, to enhance the robustness of our findings, we incorporated a comprehensive set of covariates encompassing sociodemographic characteristics, laboratory parameters, physical examination findings, comorbidities, and medication history, based on data availability, clinical relevance, and a priori selection criteria. Data collection involved the utilization of Structured Query Language (SQL) with PostgreSQL (version 14.2) to extract baseline characteristics of patients with sepsis. In the "Data Collection" section of the manuscript, we list all the covariates extracted from the MIMIC database in this study and explain when laboratory tests were collected. j

- it is suggested to use p-value instead of confidence alpha level for explaining the cut-off.

Reply: Thanks for your comment. We have revised this statement: A two-sided P value <0.05 was considered statistically significant.

- it is suggested to relocate table 1.

Reply: Thanks for your comment. We have streamlined Table 1 (The revised Table 1 was exhibited in manuscript).

- other limitations such as retrospective design, site of infection.

Reply: Thanks for your comment. We have acknowledged and supplemented these limitations in the Discussion section: Nonetheless, several important limitations of this study must be acknowledged. First, inherent to the retrospective cohort design, our study was susceptible to recall bias and potential incompleteness of medical records. While we adjusted for multiple known confounders in our logistic regression models, residual confounding from unmeasured variables may persist. Importantly, our findings demonstrate an association between β-blocker use and AKI risk in septic patients but cannot establish causality Second, as our data were obtained exclusively from the MIMIC database representing a single academic medical center, the generalizability of our findings to other populations with different demographic characteristics, healthcare systems, or practice patterns may be limited. Furthermore, the ICU-based nature of this dataset restricts the applicability of our results to non-critical care settings such as general wards or outpatient populations. Third, as an observational study, the exposure factors were not randomized. Although PSM was employed to minimize bias between beta-blocker groups, larger prospective randomized controlled trials are needed. Fourth, due to the availability of database, we were unable to stratify patients by infection site. Consequently, we could not assess whether the therapeutic effects of beta-blockers varied across different sources of sepsis. Future large-scale, well-designed multicenter prospective cohort studies or randomized controlled trials (RCTs) are warranted to further validate the association between β-blocker use and AKI risk in septic patients, while mechanistic animal studies are needed to elucidate the underlying biological pathways involved (line 290-line 310).

- is it possible to compare all types of beta-blockers with each other? in the current format, all of them were compared to labetolol.

Reply: Thanks for your comment. Our study mainly focused on three beta-blockers: Metoprolol, Esmolol and Labetalol. This study further investigated the association between different types of beta-blockers and the risk of SA-AKI. However, we did not observe any significant association between the use of metoprolol or esmolol and the risk of AKI in sepsis patients when compared with labetalol use. Following your suggestion, we further explored the relationship between different types of beta-blockers and AKI risk in sepsis patients, using metoprolol and esmolol as respective reference groups. The results were shown below:

Table 3 Association between the type of beta-blockers and SA-AKI after PSM

Variates OR (95%CI) P OR (95%CI) P OR (95%CI) P

Beta blockers

Labetalol 0.47 (0.15-1.53) 0.212 Ref 1.40 (0.87-2.25) 0.160

Metoprolol 0.34 (0.11-1.02) 0.054 0.71 (0.45-1.14) 0.160 Ref

Esmolol Ref 2.11 (0.65-6.81) 0.212 2.95 (0.98-8.91) 0.054

OR, odd ratio; CI: confidence interval; Ref: reference;

Adjusted age, weight, heart failure, systolic, temperature, SOFA, BUN, WBC, Chloride, loop diuretics, and eGFR.

We found that compared with sepsis patients who used esmolol or metoprolol, we still did not observe any statistically significant association between the use of different types of beta-blockers and the risk of AKI in sepsis patients (all P>0.05). We have updated Table 3 in the manuscript.

Reviewer #2: Your title is mostly clear but needs minor grammatical and stylistic refinements for correctness and readability. Here is the corrected version:

“Association Between Early Beta-Blocker Use and the Risk of Sepsis-Associated Acute Kidney Injury: A Retrospective Cohort Study Using the MIMIC-IV Database”

Method section:

-The statement about the Helsinki Declaration is unclear. You should say according to.

Reply: Thanks for your comment. We restated the statement on ethical approval in a formal and professional language: The database received approval from the institutional review boards of the Massachusetts Institute of Technology (MIT, located in Cambridge, Massachusetts, USA) and Beth Israel Deaconess Medical Center (BIDMC). The data is publicly available (in the MIMIC-IV database), therefore, the ethical approval statement and the requirement for informed consent were waived for this study. This study adhered to the strengthening the reporting of observational studies in epidemiology (STROBE) guidelines and complied with the principles of the Declaration of Helsinki (line 89-line 96).

-The definition of “baseline beta-blocker use” should be more explicit. Were beta-blockers initiated prior to ICU admission or only during ICU stay?

Reply: Thanks for your comments. We have clarified the definition of “baseline beta-blocker use”: The exposure of present study was the use of beta-blocker. Exposure to beta-blockers was defined as any form or dose of beta-blocker used within 24 hours after admission to the ICU (line 108-line 111).

The definition of beta-blocker use (any dose within 24 hours) may lead to misclassification bias. Did the study assess:

Duration of use? (Was it only a single dose or ongoing treatment?)

Dosage variations? (Could different doses have different effects?)

Different types of beta-blockers? (Selective vs. non-selective beta-blockers)

Reply: Thanks for your comments. We further extracted the administration methods of beta-blockers form the MIMIC-IV database and explored the association between the administration methods and the risk of AKI in patients with sepsis. The results were shown in below:

Administration methods Model 1 Model 2

OR (95%CI) P OR (95%CI) P

Continuous Med 1.16 (0.47-2.82) 0.749 1.33 (0.51-3.47) 0.562

Drug Push 0.74 (0.60-0.92) 0.006 0.73 (0.58-0.92) 0.007

No Ref Ref

In the MIMIC-IV database, "Continuous Med" and "Drug Push" represent two distinct medication administration methods. "Continuous Med" refers to intravenous infusion via pump devices, while "Drug Push" indicates rapid intravenous bolus administration. After adjusting for multiple covariates, our analysis revealed that sepsis patients receiving beta-blockers via Drug Push administration demonstrated a reduced risk of AKI compared to those not receiving beta-blockers; no statistically significant association was observed between Continuous Med administration of beta-blockers and AKI risk among sepsis patients.

This study focused on three beta-blockers: labetalol, metoprolol, and esmolol. Among these, labetalol is a non-selective beta-blocker, metoprolol and esmolol are selective beta-blockers. Additionally, due to data availability limitations in the MIMIC-IV database, the administered doses of these beta-blockers were not recorded. We acknowledge this as a study limitation and have addressed it in the discussion section.

The caliper width is set at 0.018. What is the rationale for this specific choice?

Reply: Thanks for your comment This study finally included 4,419 patients with complete clinical data of sepsis from the MIMIC-IV database. After obtaining a large research sample, we controlled for observable confounding variables through PSM to reduce selection bias and improve the balance between the comparison groups, in order to enhance the credibility of the association between the early use of beta-blockers and the risk of AKI in sepsis patients. We considered the setting of the caliper value to improve the matching accuracy. To do so, we reduced the caliper value from the commonly used 0.02 to 0.018. The sample size using beta-blockers showed a lower loss rate (750 vs 726, loss rate: 3.2%) before and after PSM. The caliper value set in this study achieved a higher sample retention and matching accuracy while ensuring the balance between the comparison groups, which to some extent improved the credibility of the association between the use of beta-blockers and the risk of AKI in sepsis patients.

Unclear Direction of Association Before PSM vs. After PSM:

Before PSM, no significant association was found between early beta-blocker use and SA-AKI.

After PSM, early beta-blocker use was associated with a lower risk of SA-AKI.

However, the interpretation of this shift is missing. The authors should discuss possible reasons why the association appears only after matching.

Reply: We appreciate the reviewer’s insightful comment. The shift in association before and after PSM likely reflects the following (line 285-line 309):

Possible unrelated causes before PSM

1. The masking effect of confounding factors: Before PSM, significant differences existed between beta-blocker users and non-users in demographic characteristics and comorbidities. These confounding factors may be associated with both beta-blockers use and the risk of AKI in patients with sepsis but exert opposing effects, thereby obscuring the true association between the early beta-blocker use and AKI in this population.

2. The unmatched raw sample may have been subject to non-random allocation, where clinicians were more likely to prescribe beta-blockers to specific subgroups of sepsis patients, thereby introducing selection bias into the sample.

3. Before PSM, the sample sizes between groups were imbalanced, with significantly fewer patients in the early beta-blocker use group compared to the non-use group (3,669 vs. 750), resulting in insufficient statistical power.

Potential explanations for the statistically significant association between early beta-blocker use and the risk of AKI in patients with sepsis after PSM include:

1. PSM attenuated the interference of confounding factors on the exposure-outcome relationship by balancing covariates between the two groups (such as age, comorbidities, SOFA score, renal function parameters, and concomitant medications), thereby unmasking the inherent protective effect of beta-blockers.

2. During the PSM process, extreme “unmatchable” cases were excluded, as these patients might have otherwise confounded the assessment of the association between early beta-blockers use and the risk of AKI in sepsis patients.

3. Moreover, clinical studies have supported the beneficial role of beta-blockers in maintain renal function.

---

## [Decision Letter · Decision Letter 1]

Association between the early use of beta-blocker and the risk of sepsis-associated acute kidney injury: A retrospective cohort study using the MIMIC-IV database

PONE-D-24-60039R1

Dear Dr. Wang,

We’re pleased to inform you that your manuscript has been judged scientifically suitable for publication and will be formally accepted for publication once it meets all outstanding technical requirements.

Kind regards,

Amirmohammad Khalaji

Academic Editor

PLOS ONE

Additional Editor Comments (optional):

Reviewers' comments:

Reviewer's Responses to Questions

**Comments to the Author**

1. If the authors have adequately addressed your comments raised in a previous round of review and you feel that this manuscript is now acceptable for publication, you may indicate that here to bypass the “Comments to the Author” section, enter your conflict of interest statement in the “Confidential to Editor” section, and submit your "Accept" recommendation.

Reviewer #1: (No Response)

2. Is the manuscript technically sound, and do the data support the conclusions?

Reviewer #1: (No Response)

3. Has the statistical analysis been performed appropriately and rigorously? 

Reviewer #1: (No Response)

4. Have the authors made all data underlying the findings in their manuscript fully available?

Reviewer #1: (No Response)

5. Is the manuscript presented in an intelligible fashion and written in standard English?

Reviewer #1: (No Response)

6. Review Comments to the Author

Reviewer #1: (No Response)

7. PLOS authors have the option to publish the peer review history of their article (what does this mean? ). If published, this will include your full peer review and any attached files.

**Do you want your identity to be public for this peer review?** For information about this choice, including consent withdrawal, please see our Privacy Policy .

Reviewer #1: No

---

## [Editor Report · Acceptance letter]

PONE-D-24-60039R1

PLOS ONE

Dear Dr. Wang,

I'm pleased to inform you that your manuscript has been deemed suitable for publication in PLOS ONE. Congratulations! Your manuscript is now being handed over to our production team.

Kind regards,

on behalf of

Dr. Amirmohammad Khalaji

Academic Editor

PLOS ONE